# Teaching Others is Teaching Yourself Regularization For Controllable Language Models

## Abstract

Large-scale pre-trained language models have achieved great success on natural language generation tasks. However, it is difficult to control the pre-trained language models to generate sentences with the expected attribute such as topic and sentiment. Recent efforts (Yang & Klein, 2021; Krause et al., 2021; Dathathri et al., 2019) on controllable language generation employ an additional attribute classifier, which guides the generation of large-scale pre-trained language models, have been shown to be efficient in controllable language generation. These methods are named *classifier-guided language models* (CGLMs). However, we find that the probabilities predicted by the attribute classifiers usually approaches $0$ or $1$, which makes it hard to distinguish sentences with different matching degrees to the expected attribute. The problem is named *the biased probability distribution* (BPD) problem. To address the problem, we investigate different methods for adjusting probability distribution and propose a *Teaching Others is Teaching Yourself* (TOTY) regularization method to smooth the probability distribution. Experiments on sentiment control and topic control tasks show that CGLMs can get better performance with guiding classifiers trained with TOTY.

## 1 Introduction

Recently, with the advances in large-scale pre-trained language model (PLM) (Radford et al., 2017; 2018; 2019; Brown et al., 2020), great progress has been made on natural language generation tasks. With billions or even trillions of parameters, and abundant unlabeled training data, PLMs can generate diverse and realistic sentences. Formally, autoregressive PLM models the probability distribution of text $X = \{x_1, x_2, ..., x_T\}$ with the chain rule:

$$p(X) = \prod_{i=1}^{T} p(x_i|x_1, x_2, ..., x_{i-1}). \tag{1}$$

However, those models are usually trained on general purpose corpus and the sentences generated by those PLMs are usually inconsistent with task requirements. Therefore, Controllable Language Generation (CLG), which aims to generate sentences that meet the requirements, has become more important in natural language generation. Controllable language generation attempts to model $p(X|a)$ where $a$ is a desired attribute (e.g. topic, length and sentiment):

$$p(X|a) = \prod_{i=1}^{T} p(x_i|X_{1:i-1}, a). \tag{2}$$

To simplify the expression, we use $X_{1:i}$ to denote the sequence $\{x_1, x_2, ..., x_i\}$.

It has been found that using an attribute classifier to guide the generation of PLMs was an efficient approach to control the PLMs to generate sentences with expected attributes (Dathathri et al., 2019; Krause et al., 2021; Yang & Klein, 2021). These methods are called classifier-guided language models (CGLMs). In CGLMs, the conditional probability at each generation step is calculated by the Bayes Rule:

$$p(x_i|X_{1:i-1}, a) \propto p(a|X_{1:i})p(x_i|X_{1:i-1}). \tag{3}$$

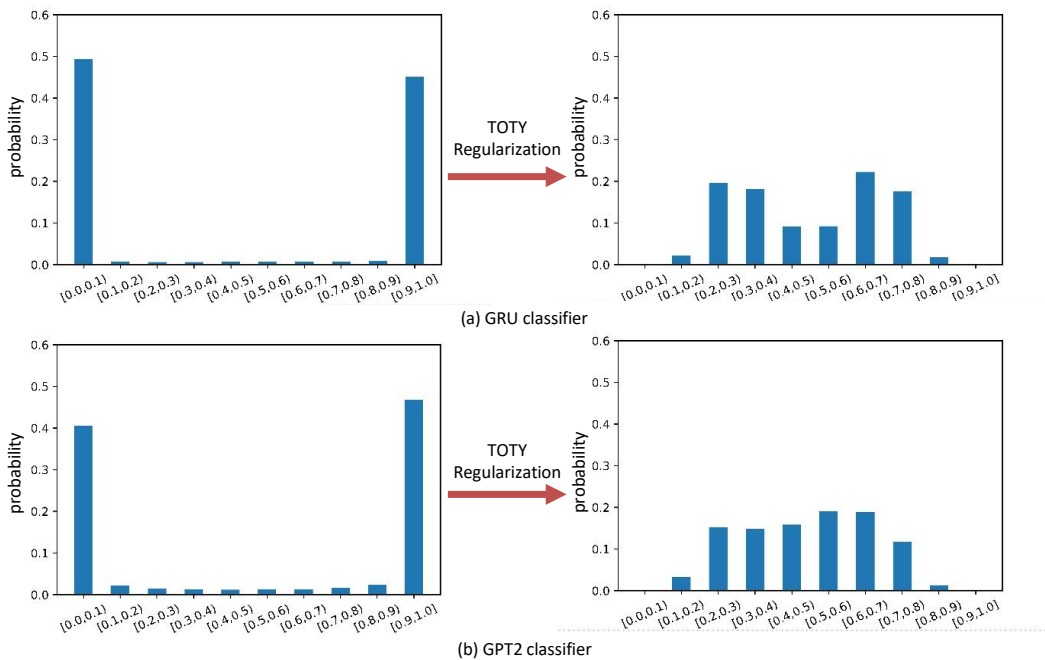

Figure 1: The probability distribution of positive sentiment predicted by different classifiers trained on the IMDB dataset. (a) GRU classifiers trained without/with TOTY regularization. (b) GPT2 classifiers trained without/with TOTY regularization.

In the formula above, $p(x_i|X_{1:i-1})$ is the unconditional probability of generating $x_i$ at step $i$, which is usually instantiated by a large-scale language model such as GPT. $p(a|X_{1:i})$ is the attribute probability that the generation result contains the attribute $a$ when starting with the $X_{1:i}$. CGLMs usually use the output an attribute classifier, known as *guiding classifier*, to model $p(a|X_{1:i})$. However, in experiments, we found that the probability distribution of $p(a|X_{1:i})$ predicted by guiding classifiers was usually very biased. To be specific, for most of sentences $X_{1:i}$, the probability predicted by guiding classifiers either approaches $0$ or approaches $1$. We call this phenomenon *the biased probability distribution* (BPD) problem.

In classification tasks, the BPD problem has little influence since these tasks only need to pick out the class with the highest probability. In other words, the concrete value of probability does not have a direct impact on classification accuracy. However, in CGLMs, the attribute of many sentences could be ambiguous. Especially for autoregressive models, the tokens are generated one after another, so the classifiers need to predict the attribute probability of many incomplete sentences. Obviously assigning a probability approaching to $0$ or $1$ to these sentences is unreasonable. For example, for the following sentences, the probabilities of positive sentiment predicted by the GRU classifier trained on the IMDB dataset (Maas et al., 2011) for a), b), and c) was 89.5%, 98.5%, 99.9%, respectively. However, only c) has a clear positive sentiment, while a) and b) do not have a clear sentiment.

  a) This tale takes place in
  b) This tale takes place in the Namib Desert of Africa.
  c) This impressive tale takes place in the Namib Desert of Africa.

A good guiding classifier for CGLMs should distinguish sentences with different matching degrees to the expected attribute, meaning that we should smooth the probability distribution predicted by the classifier. The existing methods (Wang et al., 2021; Gupta & Ramdas, 2021; Platt, 2000; Wei et al., 2022; Zadrozny & Elkan, 2001; Szegedy et al., 2016; Müller et al., 2019) for adjusting the probability distribution predicted by classifiers are usually used to address the mismatch between a model's confidence and its correctness. It will be shown that these methods does not significantly smooth the probability distribution.

In this work, we propose a simple regularization method named *Teaching Others is Teaching Yourself* (TOTY) to address the BPD problem. In TOTY, we have two classifiers for the same classification task, one is named the "teacher" and the other is the "student". The teacher classifier learns from the ground truth and teaches the student classifier. Different from knowledge distillation in which knowledge only flows from the teacher to the student, in TOTY, we align the teacher and the student together, such that they can learn from each other. Figure 1 demonstrates the probability distribution of positive sentiment predicted by classifiers trained on the IMDB dataset (Maas et al., 2011) without TOTY regularization and with TOTY regularization.

Experiments show that TOTY works well on smoothing the probability distribution of classifiers, and significantly improves the performance of CGLMs. Moreover, TOTY is an easily applicable method since it does not require complicated designing of model structure or training scheme.

## 2 RELATED WORK

### 2.1 MODELS FOR CONTROLLABLE LANGUAGE GENERATION

Controllable language models can be classified into two categories: class-conditional language models (CCLMs) and classifier-guided language models (CGLMs).

Given an expected attribute $a$, CCLM is the most straightforward approach in controllable language generation which directly models the probability $p(X|a)$ for generating sentence $X$. There are various methods to implement CCLM, such as training conditional generative models by concatenating the expected attribute to inputs (Ficler & Goldberg, 2017; Kikuchi et al., 2016), training generative adversarial networks (Yu et al., 2017), and training variational auto-encoders (Hu et al., 2017). In recent years, with the development of large-scale pre-trained language models (Radford et al., 2017; 2018; 2019; Brown et al., 2020) and prompt methods (Jiang et al., 2020; Li & Liang, 2021; Shin et al., 2020), great progress has been made in CCLMs. CTRL (Keskar et al., 2019) builds a large controllable language model trained on a large-scale corpus with 55 different control codes. Following CTRL, COCON (Chan et al., 2020) proposes three self-supervised learning loss to enhance controllability. Moreover, ACB (Yu et al., 2021) tries to disentangle the irrelevant attributes in corpus and introduces prefix-tuning in CCLM to avoid tuning a large number of parameters.

Since the training data for specific attributes is limited, CCLMs usually lead to corpus overfitting, meaning that the generated sentences of CCLMs usually highly resemble sentences in the training corpus. For example, sentences sampled from CCLMs with sentiment control trained on IMDB (Maas et al., 2011), a dataset of movie reviews for sentiment classification, usually mention words in movie reviews (e.g. movie, film, and character). However, the attribute we wish CCLMs to learn is the sentiment in IMDB rather than the context relevance of movie reviews.

CGLMs attempt to find the generation path that matches the expected attribute by using an external attribute classifier. Instead of modeling $p(X|a)$ directly, CGLMs sample $X$ according to the Bayes Rule: $p(X|a) \propto p(a|X)p(X)$. PPLM (Dathathri et al., 2019) proposes an iterative method that updates the hidden states of a language model by backpropagating gradients of an attribute model to maximize the likelihood of the expected attribute. Since PPLM requires iterations of forward and backward processes in inference, it is computationally intensive. GeDi (Krause et al., 2021) uses smaller language models to model $p(a|X)$. FUDGE (Yang & Klein, 2021) trains a discriminator to distinguish whether the desired attribute will be true in the future and use the discriminator to model $p(a|X)$. However, as discussed in Section 1, the performance of CGLMs is heavily influenced by the probability distribution of the external classifiers.

### 2.2 METHODS FOR ADJUSTING PROBABILITY DISTRIBUTION PREDICTED BY CLASSIFIERS

Most of previous methods for adjusting probability distribution predicted by classifiers are proposed to address the miscalibration problem - the mismatch between a mismatch between a model's confidence and its correctness. Miscalibration makes the predictions of deep neural networks hard to rely on and leads to the overconfidence problem. Some classical works address the problem by post-hoc methods, such as Platt Scaling (Platt, 2000) and Histogram Binning (Zadrozny & Elkan, 2001). Guo et al. (2017) systematically analysed the previous post-hoc methods. Following these works, many post-hoc methods were proposed (Kull et al., 2019; Rahimi et al., 2020; Gupta et al., 2020). Another

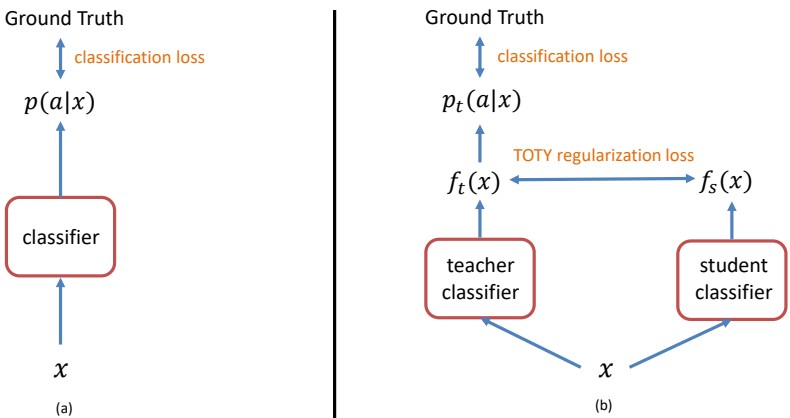

Figure 2: (a) The sketch of standard classifier training process. (b) The sketch of classifiers trained with TOTY regularization.

series of methods propose to adjusting the probability distribution by modifying the training process. Such as label smoothing (Szegedy et al., 2016; Müller et al., 2019), focal loss (Lin et al., 2017; Mukhoti et al., 2020), and LogitNorm (Wei et al., 2022).

## 3 APPROACH

### 3.1 TOTY REGULARIZATION

In this Section we introduce the TOTY regularization method. Since TOTY is not restricted to train the classifiers of CGLMs, we introduce the TOTY regularization in a general classification task. Different from classical training methods, TOTY contains two separate classifiers: the "teacher" and the "student". Figure 2 (a) and (b) demonstrates the training sketch of classical training method and TOTY respectively. The two classifiers has same input and output in training.

Let $u$ be the input of the classification task, and $y$ be the label of $u$. The TOTY regularization is defined as the euclidean distance between the logits output of the teacher and the student:

$$L_{\text{TOTY}}(u, \theta_{\text{t}}, \theta_{\text{s}}) = ||f_{\text{t}}(u, \theta_{\text{t}}) - f_{\text{s}}(u, \theta_{\text{s}})||^2, \tag{4}$$

where $f_{\text{t}}(u, \theta_{\text{t}})$ and $f_{\text{s}}(u, \theta_{\text{s}})$ denote the logit output of the teacher and the student respectively, and $\theta_{\text{t}}$ and $\theta_{\text{s}}$ are the parameters of the teacher and the student respectively. The student classifier only learns from $L_{\text{TOTY}}$:

$$L_{\text{student}} = L_{\text{TOTY}}(u, \theta_{\text{t}}, \theta_{\text{s}}). \tag{5}$$

For the teacher classifier, besides the $L_{\text{TOTY}}$, it also learns from the normal loss $L_{\text{normal}}(u, y, \theta_{\text{t}})$ for classification in classical training methods. The forms of the loss for classification in different models are various. We do not modify their own losses. Overall, for the teacher,

$$L_{\text{teacher}} = L_{\text{normal}}(u, y, \theta_{\text{t}}) + L_{\text{TOTY}}(u, \theta_{\text{t}}, \theta_{\text{s}}), \tag{6}$$

### 3.2 THEORETICAL ANALYSIS OF TOTY REGULARIZATION

In this section, we analyse how TOTY regularization smooths the probability distribution. To simplify the problem, we treat the classification task as a binary classification task. In this case, the logit output $f_{\text{t}}(u, \theta_{\text{t}})$ and $f_{\text{s}}(u, \theta_{\text{s}})$ would be scalars, and $p_{\text{t}}(y|u)$ is the sigmoid output of $f_{\text{t}}(u, \theta_{\text{t}})$. And we use the cross entropy loss as the normal loss:

$$L_{\text{normal}} = -\ln p_{\text{t}}(y|u)$$
$$= -\ln \frac{1}{1 + \exp\left(-(f_{\text{t}}(u, \theta_{\text{t}}))\right)}. \tag{7}$$

For a classifier trained with only the normal loss, ideally the training converges when

$$\frac{\partial L_{\texttt{normal}}}{\partial \theta_{\texttt{t}}} = -(1 - p_{\texttt{t}}(y|u)))\frac{\partial f_{\texttt{t}}(u, \theta_{\texttt{t}})}{\partial \theta_{\texttt{t}}} = 0. \tag{8}$$

Since the partial derivative of $f_{\texttt{t}}(u, \theta_{\texttt{t}})$ with respect to $\theta_{\texttt{t}}$ is usually not zero, obviously $p_{\texttt{t}}(y|u)$ approaches 1 when it converges.

For the teacher classifier, the gradient of the TOTY loss can be formulated as the follows:

$$\begin{aligned}\frac{\partial L_{\texttt{teacher}}}{\partial \theta_{\texttt{t}}} =& \frac{\partial L_{\texttt{normal}}}{\partial \theta_{\texttt{t}}} + \frac{\partial L_{\texttt{TOTY}}}{\partial \theta_{\texttt{t}}} \\ =& (2f_{\texttt{t}}(u, \theta_{\texttt{t}}) - 2f_{\texttt{s}}(u, \theta_{\texttt{s}}) - (1 - p_{\texttt{t}}(y|u)))\frac{\partial f_{\texttt{t}}(u, \theta_{\texttt{t}})}{\partial \theta_{\texttt{t}}}. \end{aligned} \tag{9}$$

Ideally, with TOTY regularization, when the training of the teacher classifier converges, the gradient of $L_{\texttt{teacher}}$ would approach zero, such that

$$(2f_{\texttt{t}}(u, \theta_{\texttt{t}}) - 2f_{\texttt{s}}(u, \theta_{\texttt{s}}) - (1 - p_{\texttt{t}}(y|u)))\frac{\partial f_{\texttt{t}}(u, \theta_{\texttt{t}})}{\partial \theta_{\texttt{t}}} = 0. \tag{10}$$

When the training converges, we get

$$p_{\texttt{t}}(y|u) = 1 - 2(f_{\texttt{t}}(u, \theta_{\texttt{t}}) - f_{\texttt{s}}(u, \theta_{\texttt{s}})). \tag{11}$$

Since the alignment of the teacher classifier and the student classifier would have a fitting deviation $\epsilon = f_{\texttt{t}}(u, \theta_{\texttt{t}}) - f_{\texttt{s}}(u, \theta_{\texttt{s}})$, $p_{\texttt{t}}(y|u)$ would converge to a value less than 1 when $\epsilon > 0$. So, after convergence, with the TOTY regularization the value range of $p_{\texttt{t}}(y|u)$ becomes wider, which smooths the probability distribution.

### 3.3 Applying TOTY To CGLMs

In training classifiers for guiding CGLMs, the input $u$ is a complete sentence $X_{1:T}$, and the label $y$ is the attribute $a$ of $X_{1:T}$. Since in inference the classifiers need to classify the incomplete sentence $X_{1:i}$, we apply the TOTY regularization for every incomplete sentence $X_{1:i}$ where $i$ ranges from 1 to $T$. So the TOTY regularization for classifiers of CGLMs is

$$L_{\texttt{TOTY}} = \frac{1}{T}\sum_{i=1}^{T} ||f_{\texttt{t}}(X_{1:i}, \theta_{\texttt{t}}) - f_{\texttt{s}}(X_{1:i}, \theta_{\texttt{s}})||^2. \tag{12}$$

As for the normal classification loss, we adopt the FUDGE (Yang & Klein, 2021) loss:

$$L_{\texttt{normal}} = -\frac{1}{T}\sum_{i=1}^{T}\ln p_{\texttt{t}}(a|X_{1:i}). \tag{13}$$

## 4 Experiments

To evaluate our approach, we trained TOTY with various pairs of classifiers as the teacher and the student, then tested the performance of CGLMs with these classifiers as the guiding classifier. We experimented on two controllable generation tasks: sentiment control and topic control.

### 4.1 Datasets

**IMDB:** We used the IMDB dataset (Maas et al., 2011) for sentiment control. It contains 50k samples of movie reviews labeled with sentiment tag (25k for positive sentiment and 25k for negative sentiment). We randomly chose 1k positive samples and 1k negative samples as the test set of sentiment classification. We trained our model on both positive attribute and negative attribute. In evaluation, we used the same 15 prefixes (See Section A) of sentiment control from the prior work (Dathathri et al., 2019) and generated samples started with these prefixes.

**AG News:** We used the AG News dataset (Zhang et al., 2015) for topic control. The AG News dataset contains 120k samples of news articles. We randomly chose 500 samples from each class as the test

set of sentiment classification.The samples are classified into one of the 4 classes: worlds, sports, business, and science. We experimented on all of the 4 classes. In evaluation, we used the same 20 prefixes (See Section A) of topic control from the prior work (Dathathri et al., 2019) and generated samples from these prefixes.

## 4.2 GUIDING CLASSIFIERS

We adopted three different guiding classifiers in our experiments. **GRU classifier** is a randomly initialized GRU (Cho et al., 2014) model trained on the appropriate task. **GPT2 classifier** is a GPT2 model with a one-layer classification head on top. The GPT2 was initialized with the GPT2-large implemented in Huggingface Transformers[1]. Since applying the GPT2 classifier as the guiding classifier of CGLMs would have a huge computational cost in inference, this classifier is only used as the student in our experiments. We will have more discussion on the computational cost of CGLMs guided by the GPT2 classifier in Section C. **GPT2-token classifier** is the model introduced in Section C.

## 4.3 IMPLEMENTATION DETAILS

For CGLMs, a pre-trained language model is needed for computing the unconditional generation probability $p(x_i|X_{1:i-1})$. We used the GPT2-large (Wolf et al., 2020) implemented in Huggingface Transformers[2] as the pre-trained language model in all experiments.

In training, the AdamW optimizer (Loshchilov & Hutter, 2017) was adopted in all experiments. The learning rate of AdamW was set to $5 \times 10^{-5}$. We used a batch size 64 and trained the models for 100 epochs. On average, on a Nvidia A100 Tensor Core GPU machine, each epoch took 7 minutes and 8 minutes for sentiment control and topic control respectively.

In inference, we adopted an attribute-driven nucleus sampling as the decoding strategy in all experiments. The attribute-driven nucleus sampling will be introduced in Section B.

## 4.4 EVALUATION METRICS

In controllable language generation, we need to evaluate both the attribute relevance and the linguistic quality of the generated sentences. In evaluation, for each model, we generated 50 samples per prefix and report the average score over the samples.

**Attribute relevance (AR):** For both tasks, we used classifiers based on BERT (Devlin et al., 2019) to measure the consistency between the generated sentences and the desired attribute. The BERT was initialized with the BERT-large implemented in Huggingface Transformers[3] and fine-tuned on the task-specific corpus. **Perplexity (PPL):** Since we used GPT2-large as the language model of CGLMs, for the sake of fairness, we measured the linguistic quality of generated sentences by the average perplexity of a GPT2-XL[4].

## 4.5 PROBABILITY DISTRIBUTION ANALYSIS

To further analyse the effect of TOTY, we plot the probability distribution predicted by GRU classifiers with different regularization methods trained on the IMDB dataset and the AG News dataset in Figures 3 and 4 respectively.

To simplify the expression, in the following we use TOTY($x$) to denote the TOTY regularization with model $x$ as the student. For the probability of sentences in IMDB dataset, as can be seen from Figure 3 (a), the vanilla GRU classifier has very high peaks at interval [0.0,0.1] and [0.9,1]. According to Figure 3 (b), with label smoothing, the peaks of distribution shifted, but it did not smooth the distribution. The distance that the peaks shifted is similar to the hyper-parameter $\alpha$ in label smoothing. Figure 3 (c) shows that the focal loss did not significantly alter the probability distribution, since

---

[1]https://huggingface.co/gpt2-large
[2]https://huggingface.co/gpt2-large
[3]https://huggingface.co/bert-large-uncased
[4]https://huggingface.co/gpt2-xl

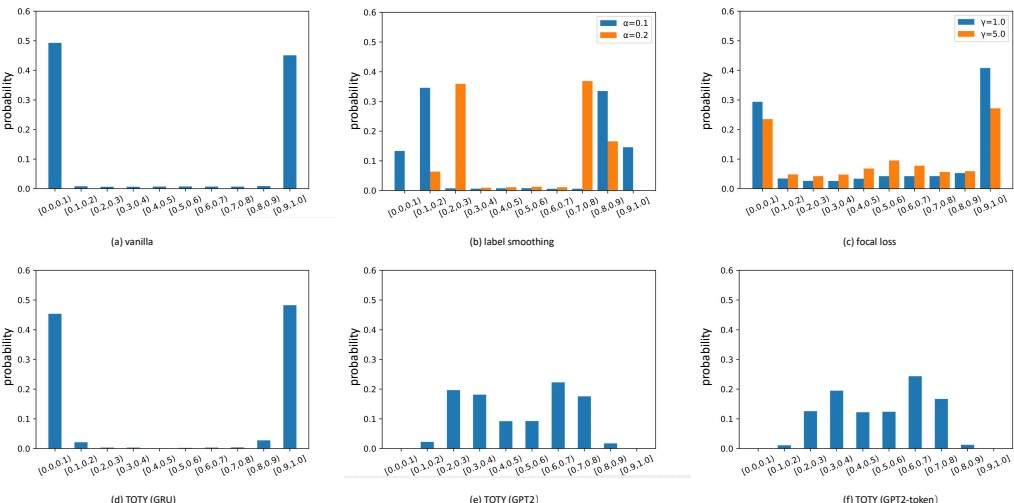

Figure 3: The probability distributions predicted by GRU classifiers with different regularization methods trained on the IMDB dataset. (a) Vanilla. (b) Label smoothing. (c) Focal loss. (d) TOTY (student: GRU classifier). (e) TOTY (student: GPT2 classifier). (f) TOTY (student: GPT2-token classifier).

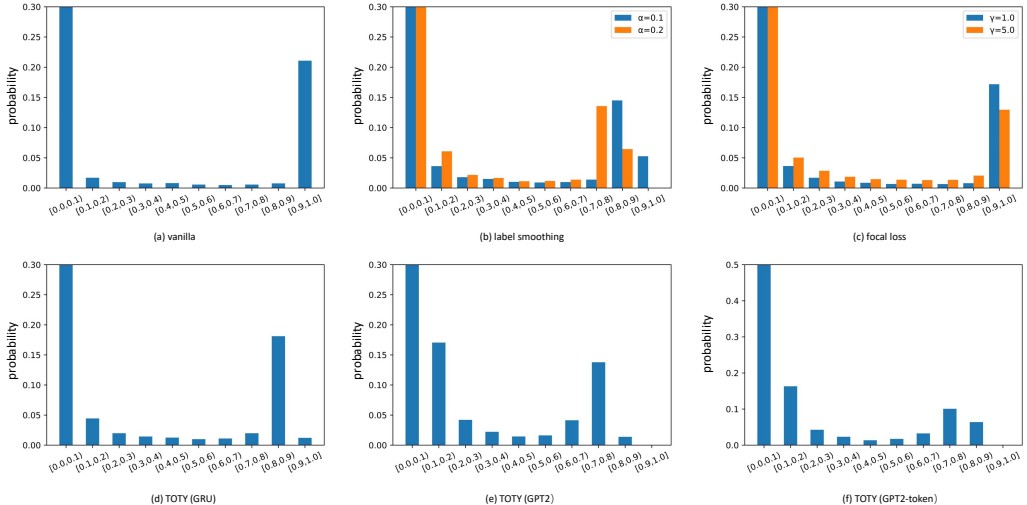

Figure 4: The probability distributions predicted by GRU classifiers with different regularization methods trained on the AG News dataset. The y-axis is truncated at 0.3 to detailed display the distribution around the secondary peak. (a) Vanilla. (b) Label smoothing. (c) Focal loss. (d) TOTY (student: GRU classifier). (e) TOTY (student: GPT2 classifier). (f) TOTY (student: GPT2-token classifier).

the probability distribution predicted by models with focal loss still had very high peaks at [0.0,0.1] and [0.9,1]. Figure 3 (d), (e), and (f) show the probability distributions of classifiers trained with TOTY by using different models as the student. It is shown that with TOTY(GRU), the probability distribution predicted by the teacher GRU classifier was not well smoothed, while with TOTY(GPT2) and TOTY(GPT2-token), the probability distribution got much smoother. We think this phenomenon can be explained with the theoretical analysis in Section 3.2. As we discussed in Section 3.2, the probability distribution will get smoother when the fitting deviation $\epsilon$ gets larger. Compared with aligning models with different structures, the absolute value of $\epsilon$ is small when the teacher and the student had the same structure, such that the value range of probability didn't get wider and the probability could not be well smoothed.

Table 1: Results on sentiment control and topic control task. LS and Focal denotes the label smoothing and focal loss method respectively. TOTY($x$) means that the student model is $x$.

| Classifier | Regularization | Negative Sentiment | | Positive Sentiment | | Topic | |
|---|---|---|---|---|---|---|---|
| | | AR(%)↑ | PPL↓ | AR(%)↑ | PPL↓ | AR(%)↑ | PPL↓ |
| GRU | vanilla | 90.3 | 33.6 | 98.8 | 72.1 | 97.7 | 33.0 |
| | LS ($\alpha = 0.1$) | 89.7 | 32.0 | 98.7 | 37.0 | 96.9 | 33.0 |
| | LS ($\alpha = 0.2$) | 84.4 | 28.1 | 96.8 | 34.2 | 96.2 | 31.9 |
| | Focal ($\gamma = 1.0$) | 86.7 | 34.6 | 98.3 | 57.0 | 96.6 | 31.5 |
| | Focal ($\gamma = 5.0$) | 86.1 | 35.1 | 96.5 | 67.6 | 91.4 | 32.3 |
| | TOTY (GRU) | 88.0 | 32.0 | 98.3 | 47.0 | 97.8 | 31.0 |
| | TOTY (GPT2) | 93.7 | 35.4 | 99.2 | **26.6** | 97.8 | **27.2** |
| | TOTY (GPT2-token) | **93.9** | **26.5** | **99.3** | 36.7 | **98.5** | 29.1 |
| GPT2-token | vanilla | 92.5 | 28.7 | 98.4 | 25.9 | 98.3 | 22.9 |
| | LS ($\alpha = 0.1$) | 58.4 | 61.6 | 91.7 | 39.9 | 98.5 | 22.7 |
| | LS ($\alpha = 0.2$) | 60.0 | 54.4 | 88.1 | 40.7 | 97.4 | 22.2 |
| | Focal ($\gamma = 1.0$) | 60.1 | 36.2 | 90.4 | 27.2 | 25.5 | 26.8 |
| | Focal ($\gamma = 5.0$) | 65.6 | 34.7 | 91.6 | 30.7 | 90.9 | 43.0 |
| | TOTY (GRU) | 94.4 | 24.3 | 99.4 | 26.7 | 99.0 | 22.2 |
| | TOTY (GPT2) | **94.9** | **24.0** | 98.8 | **18.4** | 98.5 | **16.9** |
| | TOTY (GPT2-token) | 92.0 | 25.1 | **99.5** | 21.1 | **99.3** | 19.9 |

For the probability of sentences in the AG News dataset, since the dataset is a 4-classes classification task and we only compute the probability distribution of one of the classes, it's a normal phenomenon that there is a high peak of distribution at interval [0.0,0.1). So, in Figure 4, we only care about the secondary peak in the distribution. From Figure 4, we can draw similar conclusion with the conclusion of Figure 3, such as the label smoothing only shifted the secondary peak and the focal loss did not significantly alter the probability distribution. From Table 1, we find that on the topic control, the AR of CGLM with GRU classifier trained by TOTY(GPT2-token) is better than trained by TOTY(GRU) and TOTY(GPT2). This result can be explained from the probability distribution of these models since compared to the Figure 4 (d) and (e) the distribution in Figure 4(f) has lower secondary peak.

## 4.6 EXPERIMENTAL RESULTS

In our experiments, we used the vanilla classifier trained with the classification loss in equation 13 without any regularization method as the baseline model. To verify the effect of TOTY on CGLMs, we compared classifiers trained with TOTY with two previous regularization methods for adjusting probability distribution: **label smoothing** (Szegedy et al., 2016; Müller et al., 2019) and **focal loss** (Lin et al., 2017). In experiments of label smoothing, we tested the hyper-parameter $\alpha = 0.1$ and $\alpha = 0.2$ separately. And in experiments of focal loss, we tested the hyper-parameter $\gamma = 1.0$ and $\gamma = 5.0$ separately.

Table 1 shows the results on sentiment control and topic control. In most of experiments, compared to the vanilla model, label smoothing and focal loss could not improve the performance on AR or PPL, and in some cases (such as the GRU classifier with label smoothing $\alpha = 0.2$ on negative sentiment control, and the GPT2-token classifier with focal loss $\gamma = 1.0$ on topic control) these regularization methods even seriously harmed the performance.

And compared with the baseline models, in most of experiments, TOTY significantly improved the attribute relevance with negative sentiment, positive sentiment and expected topic, meanwhile reduced the perplexity metric.

However, we also found the GRU classifier for sentiment control when the student was a GRU classifier had no improvement. Comparing the results with the probability distribution in Figure 3, we conclude that whether TOTY works well on smoothing the probability distribution determines the performance of CGLMs with the classifiers as the guiding classifier.

Table 2: The stepwise probabilities assessed by different classifiers. The data in each step stands for the probability of the desired attribute for the sentence from the beginning to the current step.

| Positive Sentiment | The | food | is | awful | but | the | service | is | nice |
|---|---|---|---|---|---|---|---|---|---|
| GRU-vanilla | 0.50 | 0.43 | 0.41 | 0.02 | 0.05 | 0.08 | 0.09 | 0.10 | 0.02 |
| GRU-TOTY(GRU) | 0.54 | 0.47 | 0.49 | 0.01 | 0.01 | 0.01 | 0.01 | 0.01 | 0.01 |
| GRU-TOTY(GPT2) | 0.51 | 0.48 | 0.49 | 0.22 | 0.33 | 0.34 | 0.33 | 0.36 | 0.46 |
| GRU-TOTY(GPT2-token) | 0.51 | 0.50 | 0.52 | 0.19 | 0.29 | 0.35 | 0.34 | 0.37 | 0.44 |
| Sports Topic | The | Italian | athlete | attended | the | World | War | in | 1940 |
| GRU-vanilla | 0.22 | 0.22 | 0.94 | 0.97 | 0.99 | 1.00 | 1.00 | 1.00 | 1.00 |
| GRU-TOTY(GRU) | 0.24 | 0.30 | 0.80 | 0.88 | 0.89 | 0.89 | 0.86 | 0.87 | 0.86 |
| GRU-TOTY(GPT2) | 0.24 | 0.24 | 0.63 | 0.64 | 0.67 | 0.68 | 0.58 | 0.61 | 0.54 |
| GRU-TOTY(GPT2-token) | 0.23 | 0.29 | 0.78 | 0.72 | 0.73 | 0.77 | 0.62 | 0.55 | 0.49 |

### 4.7 STEPWISE ANALYSIS OF CLASSIFIERS WITH TOTY

Table 2 demonstrates the stepwise probabilities of positive attributes and sports topics. From the stepwise probabilities, we found that the stepwise probabilities assessed by GRU classifiers with TOTY(GPT2) and TOTY(GPT2-token) flowed along with the conversion of the sentiment and topic in the sentences. For "The food is awful but the service is nice", the sentiment in the sentence converts from negative to positive, and the stepwise probabilities of positive attribute decreased firstly and then increased gradually. For "The Italian athlete attended the World War in 1940", the stepwise probabilities decreased after the word "War", which is reasonable. However, the vanilla classifier and the GRU classifier with TOTY(GRU) performed poorly on both the sentences. They classified "The food is awful but the service is nice" as a very negative sentence, and classified "The Italian athlete attended the World War in 1940" as a sentence strongly related to the sports topic. To conclude, the stepwise probability of models with biased probability distribution usually could not adapt the attribute conversion of the sentences, while the models with smoothed probability distribution performs well on attribute conversion.

## 5 CONCLUSION

In this work, we identify the BPD problem which harms the performance of CGLMs. To address the BPD problem, we introduce TOTY, an easily applicable and efficient regularization method for smoothing the probability distribution of classifiers. By aligning the teacher classifier and the student classifier together, TOTY smooths the probability distribution of classifiers. Experiments on sentiment control task and topic control task show that TOTY can significantly improve the performance of CGLMs. We also think that TOTY might be a useful regularization method for other tasks.

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

## A    PREFIXES USED IN EXPERIMENTS FOR SENTENCE GENERATION

We used the same 15 prefixes for sentiment control and 20 prefixes for topic control from PPLM.

**Prefixes for sentiment control:** "Once upon a time", "The book", "The chicken", "The city", "The country", "The horse", "The lake", "The last time","The movie", "The painting", "The pizza", "The potato", "The president of the country", "The road", "The year is 1910.".

**Prefixes for topic control:** "In summary,", "This essay discusses", "Views on", "The connection", "Foundational to this is", "To review,", "In brief,", "An illustration of", "Furthermore,", "The central theme", "To conclude,", "The key aspect", "Prior to this", "Emphasised are", "To summarise,", "The relationship", "More importantly,", "It has been shown", "The issue focused on", "In this essay".

## B    ATTRIBUTE-DRIVEN NUCLEUS SAMPLING

As we discussed in Section 1, CGLMs use the Bayes Rule in equation 3 to calculate the conditional probability. However, in inference, $x_i$ is to be generated at step $i$ in inference. To get the probability distribution of $x_i$, we need to calculate $p(w|X_{1:i-1})$ for all token $w$ in the vocabulary, such that we need to calculate $p(a|X_{1:i-1}, w)$ for all token $w$ in the vocabulary.

Following previous CGLM works (Dathathri et al., 2019; Krause et al., 2021), we applies weighted decoding in our model by introducing a non-negative hyper-parameter $\lambda$. The conditioned probability with weighted decoding is

$$p(w|X_{1:i-1}, a) \propto p(a|X_{1:i-1}, w)^{\lambda} p(w|X_{1:i-1}). \tag{14}$$

$\lambda$ is a hyper-parameter to balance controllability and fluency of generation. When $\lambda$ is $0$, the conditioned probability degrades into the unconditioned probability. In all of our experiments, $\lambda$ was set to 5.

Moreover, inspired by the decoding strategy in the original paper of GeDi (Krause et al., 2021), we design an attribute-driven nucleus sampling in inference to improve the controllability and the linguistic quality. The attribute-driven nucleus sampling contains two filters. The first filter is a standard nucleus sampling filter (Holtzman et al., 2019) acting on the unconditional probability distribution $p(w|X_{1:i-1})$ to maintain the linguistic quality of generation. We define $V_k$ as the set of $k$ tokens $w$ with the highest $p(w|X_{1:i-1})$. With a filter probability $\rho_1$, $\hat{k}(\rho_1)$ is defined as

$$\hat{k}(\rho_1) = \operatorname*{argmin}_{k}(\sum_{w \in V_k} p(w|X_{1:i-1}) \geq \rho_1). \tag{15}$$

With the first filter, we get the $\hat{k}(\rho_1)$ tokens with the highest $p(w|X_{1:i-1})$ which form a set $V_{\hat{k}(\rho_1)}$. Inspired by the decoding strategy of GeDi (Krause et al., 2021), the second filter, named "attribute-driven filter", acts on the conditional probability distribution $p(w|X_{1:i-1}, a)$, aiming at making the generated sentences well conditioned on the attribute $a$. We define $U_m$ as the set of $m$ tokens $w$ in $V_{\hat{k}(\rho_1)}$ with the highest $p(a|X_{1:i-1}, w)$. With a filter probability $\rho_2$, $\hat{m}(\rho_2)$ is defined as

$$\hat{m}(\rho_2) = \operatorname*{argmin}_{m}(\sum_{w \in U_m} p(w|X_{1:i-1}, a) \geq \rho_2). \tag{16}$$

Finally we sample from the set $U_{\hat{m}(\rho_2)}$. Experimentally, we set $\rho_1 = 0.9$, $\rho_2 = 0.3$ in all experiments.

## C GPT2-TOKEN CLASSIFIER

As we discussed in Section B, we need to calculate $p(a|X_{1:i-1}, w)$ for all token $w$ in the vocabulary. For the GPT2 classifier, we need to extract the features of the sentence $\{X_{1:i-1}, w\}$ for all token $w$ in the vocabulary. Usually the vocabulary size of GPT2 is more than 50k, and on a single NVIDIA A100 Tensor Core GPU machine, extracting the feature of a sentence by GPT2-large takes about 0.03s. So it is a huge cost for extracting the features of sentence $\{X_{1:i-1}, w\}$ for all token $w$ in the vocabulary. To reduce the computational cost, we introduce the GPT2-token classifier.

Let $h_1, ..., h_T$ denote the last hidden state of a GPT2 model with $X_{1:T}$ as the input. For each $i$, the dimension of $h_i$ is $d_h$.

Let $e_w$ denote the embedding of token $w$. The dimension of $e_w$ is $d_e$. The GPT2-token classifier $M$ receives $h_{i-1}$ and $e_w$ as the input. Different from the GPT2 classifier, GPT2-token classifier do not need to extract the feature of $\{X_{1:i-1}, w\}$ by GPT2 for all token $w$ in vocabulary, which significantly reduces the computational cost. The structure of the GPT2-token classifier is a gated neural network:

$$
\begin{aligned}
g_{i-1} &= W_g h_{i-1} + b_g, \\
r_i &= \sigma(W_{1r} e_w + W_{2r} g_{i-1} + b_r), \\
z_i &= \sigma(W_{1z} e_w + W_{2z} g_{i-1} + b_z), \\
n_i &= \text{Tanh}(W_{1n} e_w + b_{1n} + r_i(W_{2n} g_{i-1} + b_{2n})), \\
o_i &= \text{ReLU}((1 - z_i)n_i + z_i g_{i-1}), \\
M(h_{i-1}, e_w) &= W_0 o_i,
\end{aligned}
\tag{17}
$$

where $W_g$ is $d_h \times d_h$ matrix, a $W_{1r}, W_{1z}, W_{1n}$ are $d_h \times d_e$ matrices, $W_{2r}, W_{2z}, W_{2n}$ are $d_h \times d_h$ matrices, $b_g, b_r, b_z, b_{1n}$ and $b_{2n}$ are $d_h$ dimensional vectors, $W_0$ is a $d_h$ dimensional row vector and Tanh is the hyperbolic tangent function.

