# OpenReview forum: "Teaching Others is Teaching Yourself Regularization For Controllable Language Models"
_ICLR.cc/2023/Conference — Submitted to ICLR 2023_

### Official Review · Reviewer_jJwL · 2022-10-25

**Confidence:** 3
**Correctness:** 2
**Technical Novelty And Significance:** 3
**Empirical Novelty And Significance:** 1
**Recommendation:** 3

**Clarity, Quality, Novelty And Reproducibility:**

Clarity: I think the paper is written decently and can be followed.

Quality/Novelty: I think the method and idea for calibration is important, but one that's been studied in many contexts and perhaps this one. I'm not an expert in the calibration literature to comment more, but this seems like a relevant and important problem to work on for many reasons. I'd say its reasonably novel.

Reproducibility: I'm not sure I could reproduce this exactly from the descriptions in the paper.

**Strength And Weaknesses:**

Strengths:
- The method is relatively simple, although has a lot of potential confounders and moving parts (choice of architecture of the student/teacher models, hyperparameters with their training/fine-tuning etc.)

Weaknesses:
- The body of prior work on classifier-guided generation with language models in both a supervised and unsupervised perspective is vast and not compared with or discussed. For example: Plug and Play Autoencoders for Conditional Text Generation (Mai et al. 2020), DExperts: Decoding-Time Controlled Text Generation with Experts and Anti-Experts (Liu et al. 2021), and Extracting Latent Steering Vectors from Pretrained Language Models (Subramani et al. 2022) all work on sentiment control in either a supervised or unsupervised context. Quark (Lu et al. 2022) has really strong results on supervised sentiment control. There are numerous others, but these are just a few.
- It's really unclear whether the smoothing is significantly helpful. The results here seem mixed and for relatively weak generators.
- It'd be nice to see other attributes be controlled such as toxicity.



**Summary Of The Paper:**

The authors propose a method called Teaching Others is Teaching Yourself (TOTY) regularization that smoothes the probability distribution that attribute discriminators produce when guiding generation. They show that augmenting classifier-guided language models with TOTY helps for sentiment control and topic control.

**Summary Of The Review:**

I think the work is a good step towards a comprehensive study of whether the smoothing is helpful and how it affects different tiers of generators. The experimentation is quite limited and not contextualized with the broader literature/models on steering generation/controllable text generation/etc. Without a much stronger set of experiments, I cannot recommend acceptance.

---

> ### Author Response · Authors · 2022-11-18
> **Response to reviewer jJwL**
>
> Q1: The body of prior work on classifier-guided generation with language models in both a supervised and unsupervised perspective is vast and not compared with or discussed.
>
> Response: Thank you for your advice. We show the performance of some of controllable language models in other works in Table (a). And we have added the performance of these models in Table 1 in our paper.
>
> PPLM (Dathathri et al., 2019)	                           63.9	                        17.0	                        -	               -
>
> GeDi (Krause et al, 2021)	                           95.2	                        75.8	                       98.1	               76.8
>
> FUDGE (Yang & Klein, 2021) 	                           70.9	                        11.1	                       85.1	               14.5
>
> Contrastive Prefixes (Qian et al, 2022)	   81.6	                        14.1	                       91.6	               22.7
>
> DExperts	(Liu et al, 2021)                           95.3	                        42.8	                       -	               -
>
> Quark (Lu et al, 2022)	                           96.1	                        14.6	                       -	               -
>
> GRU baseline	                   94.6	                        52.9	                      97.7	               33.0
>
> GPT2-token baseline	   95.5	                        27.3	                      98.3	               22.9
>
> GRU-TOTY(GPT2-token)   96.6	                        31.6	                      98.5	               29.1
>
> GPT2-token-TOTY(GPT2) 96.9	                        21.2	                      98.5	               16.9
>
> Table (a): Performance of models in other papers and some models in our experiments.
>
> Q2: It's really unclear whether the smoothing is significantly helpful. The results here seem mixed and for relatively weak generators.
>
> Response: As shown in Table 1 in our paper, Most of TOTY pairs can get better performance compared with the vanilla models except the GRU-(GRU student) TOTY pair on sentiment control task. We guess you are concerned with the GRU-(GRU student) TOTY pair. As discussed in the second paragraph of section 4.5: this pair is used to show that when the teacher and the student use the same structure, TOTY might not successfully smooth the probability distribution. We also prove it theoretically in the second paragraph of section 4.5. When we the teacher and the student use different models, this problem can be avoided.
> Moreover, as shown in Table (a), the baseline models in our experiment are not weak and comparable with SOTA models.
>
> Q3: It'd be nice to see other attributes be controlled such as toxicity.
>
> Response: To test on the detoxification task, we trained the vanilla models and models with TOTY on the Jigsaw Toxic Comment Classification Challenge dataset . In our experiments, we used 10 very toxic prefixes as the start of generation, and let the model to detoxify these prefixes. Following our setting of sentiment control and topic control, each model generated 50 samples per prefix. The results is shown in Table (b). We can draw a similar conclusion with sentiment control: in most of the experiments, TOTY have better AR and PPL performance compared with the vanilla models except the GRU-GRU pair.
>
> model	AR(%)↑	PPL↓
>
> Original GPT2 (without control)	21.0	17.8
>
> Vanilla GRU	37.0	47.0
>
> GRU-TOTY(GRU student)	36.5	42.9
>
> GRU-TOTY(GPT2 student)	37.5	34.5
>
> GRU-TOTY(GPT2-token student)	40.0	33.9
>
> Vanilla GPT2-token	36.5	56.3
>
> GPT2-token-TOTY(GRU student)	40.5	35.9
>
> GPT2-token-TOTY(GPT2 student)	38.5	26.2
>
> GPT2-token-TOTY(GPT2-token student)	39.5	29.6
>
> Table (b): Results on detoxification.

---

### Official Review · Reviewer_eggk · 2022-10-25

**Confidence:** 4
**Correctness:** 3
**Technical Novelty And Significance:** 2
**Empirical Novelty And Significance:** 2
**Recommendation:** 3

**Clarity, Quality, Novelty And Reproducibility:**

Why is the classifier still based on GRU, but not pre-trained language models, such as RoBERTa?

**Strength And Weaknesses:**

Strength
1. The proposed method can significantly improves the performance of classifier-guided language models, and get better performance than simple label smoothing and focal loss.
2. The paper is well-written and easy to read.

Weakness
1. The baselines are a little bit weak. It would be better to compare with PPLM (Dathathri et al., 2019) and FUDGE (Yang & Klein, 2021) discussed in the related works.
2. The proposed method is actually self-distillation or may be called co-training. It's a well-known method. Not surprised to see the smoothing ability on classifier. I think the title is talking about self-distillation loss as the regularization. There may be no need to name it "TEACHING OTHERS IS TEACHING YOURSELF REGULARIZATION".

**Summary Of The Paper:**

The authors propose a simple regularization method named Teaching Others is Teaching Yourself (TOTY) to address the BPD problem. The proposed method is similar to self-distillation. Different from knowledge distillation in which knowledge only flows from the teacher to the student, in TOTY, the authors align the teacher and the student together, such that they can learn from each other. Experiments show that TOTY works well on smoothing the probability distribution of classifiers, and significantly improves the performance of classifier-guided language models.

**Summary Of The Review:**

Overall, I think the baselines are not strong enough and the proposed method is very similar to self-distillation which has been widely used in many areas.

---

> ### Author Response · Authors · 2022-11-18
> **Response to reviewer eggk**
>
> Q1: The baselines are a little bit weak. It would be better to compare with PPLM (Dathathri et al., 2019) and FUDGE (Yang & Klein, 2021) discussed in the related works.
> Response: Thank you for your advise, now we compare PPLM, FUDGE and other controllable language models in Table (a). It can be seen from table (a) that the baseline models in our experiment are comparable with SOTA models such as Quark (Lu et al, 2022) and DExperts (Liu et al, 2021) .
>
> model	                           AR (sentiment)	PPL(sentiment)	AR (topic)	PPL (topic)
>
> PPLM (Dathathri et al., 2019)	                           63.9	                        17.0	                        -	               -
>
> GeDi (Krause et al, 2021)	                           95.2	                        75.8	                       98.1	               76.8
>
> FUDGE (Yang & Klein, 2021) 	                           70.9	                        11.1	                       85.1	               14.5
>
> Contrastive Prefixes (Qian et al, 2022)	   81.6	                        14.1	                       91.6	               22.7
>
> DExperts	(Liu et al, 2021)                           95.3	                        42.8	                       -	               -
>
> Quark (Lu et al, 2022)	                           96.1	                        14.6	                       -	               -
>
> GRU baseline	                   94.6	                        52.9	                      97.7	               33.0
>
> GPT2-token baseline	   95.5	                        27.3	                      98.3	               22.9
>
> GRU-TOTY(GPT2-token)   96.6	                        31.6	                      98.5	               29.1
>
> GPT2-token-TOTY(GPT2) 96.9	                        21.2	                      98.5	               16.9
>
> Table (a): Performance of models in other papers and some models in our experiments.
>
> Q2: The proposed method is actually self-distillation or may be called co-training. It's a well-known method. Not surprised to see the smoothing ability on classifier. I think the title is talking about self-distillation loss as the regularization. There may be no need to name it "TEACHING OTHERS IS TEACHING YOURSELF REGULARIZATION"
>
> Response: Our method is quite different from self-distillation (Zhang et al, 2019). Self-distillation is proposed to boost the student model, while TOTY is proposed to smooth the probability distribution of both the teacher and the student. In self-distillation the teacher model is usually deeper than the student model, while in TOTY we don’t have the requirement, both the teacher and the student could be deeper (such as we can use the GRU model as the teacher, and GPT-2 model as the student). Moreover, in self-distillation, the supervision of teacher model only comes from labels, and supervision of the student models comes from both labels and the teacher model. But in TOTY, the supervision of teacher model comes both from labels and from the student, and the supervision of the student models comes only from the teacher. So TOTY and self-distillation are totally different.

---

### Official Review · Reviewer_absA · 2022-10-25

**Confidence:** 2
**Clarity, Quality, Novelty And Reproducibility:** 1. Although the paper shows that the …
**Correctness:** 3
**Technical Novelty And Significance:** 2
**Empirical Novelty And Significance:** 3
**Recommendation:** 5

**Strength And Weaknesses:**

The proposed method is simple and achieves better performance with the help of the regularization loss.

**Summary Of The Paper:**

This paper proposes a simple regularization method named TOTY to address the biased probability distribution problem. The proposed method introduces a teacher and student model for the classifier and employs a regularization loss to improve the classifier. The proposed method achieves better performance on IMDB and AG News.

**Summary Of The Review:**

The proposed method is simple and effective. But I am confused about the motivation, which improves the generation via smoothing the probability distributions.

---

> ### Author Response · Authors · 2022-11-18
> **Response to reviewer absA**
>
> Q1: Although the paper shows that the proposed method smooths probability distributions of classifiers and achieves better performance. I still feel confused about the motivation. Why smoothing the probability distributions is helpful?
>
> Response: In generation, the classifiers have to predict the probability of many incomplete sentences which do not have a clear attribute. However, without smoothing, these incomplete sentences are usually given a very low or very high probability because of the BPD problem. When the distribution gets smoothed, the probabilities of these incomplete sentences are more distinguishable, which helps the PLM to generate sentences that fit the expected attributes.
>
> Q2: It is also not clear to me why the standard GRU classifier has the biased probability distribution problem? It seems that a much heavier classifier GPT-2 also has the same problem.
>
> Response: Yes, GPT-2 classifier also has the BPD problem. We did not used the GPT-2 classifier as the guider classifier because it will lead to a huge computational cost. We discussed the reason of the computational cost of using GPT-2 as the guider classifier in Section C of Supplementary Material.

---

### Official Review · Reviewer_V4K8 · 2022-10-30

**Confidence:** 4
**Clarity, Quality, Novelty And Reproducibility:** Please see the review above
**Correctness:** 4
**Technical Novelty And Significance:** 4
**Empirical Novelty And Significance:** 2
**Recommendation:** 6

**Strength And Weaknesses:**

Strengths:

-- The observed problem is important to tackle.

-- The proposed regularizer is reasonable and simple to implement.

-- The proposed approach exhibits a greater amount of smoothing than "label smoothing" and "focal loss" baselines, especially with GPT-2 based students.

-- The generated text seems to be on par or better than the vanilla approach with GPT-2 student classifier.

Weaknesses:

-- In the evaluation setup for topic control, only 4 topics instead of 7 topics as explored in the prior work are considered. Justification for omission of these topics and the potential impact on the empirical comparison should be provided.

-- This paper lacks human evaluation of the generated sentences which is important because the automatic metrics, while helpful and informative, are imperfect.

-- The paper shows that GPT-2 based student is crucial for the success of the approach. However, more analysis on why this is the case would make the paper stronger.

-- Some examples of generated samples would enhance the understanding the capability of the proposed approach.

**Summary Of The Paper:**

This paper focuses on classifier guided controllable generation of text with autoregressive models. Following an observation that the stepwise classifiers tend to always be peaked, this work argues that it is an undesirable characteristic because of the loss of contrast between classifier's predictions on partial completions and fuller completions which might cause stepwise generation to be miscalibrated. The solution proposed for this problem is TOTY, which instead of training just 1 classifier at each step, trains a teacher classifier at each step and additionally, simultaneously trains a student classifier to match the teacher's logits which causes encourages teacher's predictions to be less peaked. The approach is tested on topic and sentiment control using lightweight GRU and GPT-2 based classifiers as students. It is empirically compared against other ways of smoothing/calibrating peaked distributions.

**Summary Of The Review:**

Overall, this paper identifies an important problem for classifier-guided autoregressive controlled generation and proposes a reasonable solution which is shown to beneficial empirically.

---

> ### Author Response · Authors · 2022-11-18
> **Response to Reviewer V4K8**
>
> Q1: In the evaluation setup for topic control, only 4 topics instead of 7 topics as explored in the prior work are considered. Justification for omission of these topics and the potential impact on the empirical comparison should be provided.
>
> Response: In the original paper of PPLM, the authors used a bag of keywords to define the 7 topics, which is different from our setting. We used the AG News dataset which contains 4 topics to train the attribute classifiers in our experiments. There are a lot of controllable language model works using this setting, such as “Attribute Alignment: Controlling Text Generation from Pre-trained Language Models” (Yu et al. 2021) [1] and “Controllable Natural Language Generation with Contrastive Prefixes” (Qian et al. 2022) [2].
>
> Q2: The paper shows that GPT-2 based student is crucial for the success of the approach. However, more analysis on why this is the case would make the paper stronger.
>
> Response: In our experiments, we showed that the performance improvement of TOTY came from the smoothing of probability distribution. The GRU-(GRU student) pair did not have good performance on sentiment-control task because it did not smooth the distribution (see Figure 3 (d)). GRU could also be a good student. For example, the GPT2-token-(GRU student) pair had 2.4 and 1.0 percent improvement on AR for negative and positive sentiment control respectively.
>
> Q3: Some examples of generated samples would enhance the understanding the capability of the proposed approach.
>
> Response: Thank you for your advice. We now we show some generated samples in our paper.

---

### Decision · Program_Chairs · 2023-01-20

**Decision:**

Reject

**Justification For Why Not Higher Score:**

This work proposes a simple approach to fix an important problem. However, the motivation for the approach is unclear, particularly why smooth distributions help. The experiments are also quite limited without enough comparison or discussion of existing similar approaches for controlled text generation.

**Justification For Why Not Lower Score:**

N/A

**Metareview: Summary, Strengths And Weaknesses:**

This work aims to solve the difficult problem of controlled generation. The work points out that existing classifier-guided language models have issues in that their predicted probabilities are extreme (close to 0/1) and that this can make it hard to guide the generation. The work proposes the 'Teaching Others is Teaching Yourself' TOTY regularisation method, which trains both a teacher classifier and a student classifier which prevents the teacher's predictions from being so peaked.

Strengths:
* The work tackles an important problem.
* The approach is simple to implement and easy to understand.
* The approach performs better than label smoothing and focal loss baselines.
* The paper is well-written.

Weaknesses:
* The paper doesn't have a through discussion of related approaches and the baselines are relatively weak.
* The motivation for the approach is unclear such as why or whether the smoothing is significantly helpful.
* The approach is similar to existing self-distillation methods so there may be no need for a new name.